# Overviewing the Ground Reality of Microplastic Effects on Seafoods, Including Fish, Shrimps and Crabs: Future Research Directions

**DOI:** 10.3390/foods11243976

**Published:** 2022-12-08

**Authors:** Judy Gopal, Iyyakkannu Sivanesan, Manikandan Muthu, Jae-Wook Oh

**Affiliations:** 1Department of Research and Innovation, Saveetha School of Engineering, Saveetha Institute of Medical and Technical Sciences (SIMATS), Chennai 602105, India; 2Department of Bioresources and Food Science, Konkuk University, 1, Hwayang-dong, Gwangjin-gu, Seoul 05029, Republic of Korea; 3Department of Stem Cell and Regenerative Biotechnology, Konkuk University, Seoul 05029, Republic of Korea

**Keywords:** microplastics, seafood, canned fish, regional impact, mitigation

## Abstract

While plastics are already notorious for their accumulation in the environment, which poses environmental challenges, invisible microplastics (MPS) are an even greater challenge. This review focuses on consolidating the reports available on MP accumulation in edible marine and freshwater fishes, shrimps, and crabs. The reality as to whether MPs in these edible aquatic organisms are really a cause of high concern is questioned and discussed. While the entrails of aquatic organisms are reported to contain high levels of MPs, because these products are consumed after the removal of the entrails and gut area in the majority of cases, the MP threat is questionable. The existence of MPs in these aquatic sources is validated but their potency in harming humans, aquatic organisms, and other interlinked species is unassessed. To overcome the difficulty in tracing the movement of MPs in a bigger ecosystem, this review proposes laboratory-based pilot studies mimicking real-world conditions, which will help us to understand the kinetics of MPs in the food chain. The effects of MPs on human welfare and health are yet to be assessed, and this is another gap that needs attention.

## 1. Introduction

In the last few decades, plastic production and use have increased exponentially all over the world because plastic goods have become a part of people’s daily lives. An estimated 250–300 million tons are produced annually [1], with an expanded 25-fold increase in plastics in the past 40 years as a result of their minimal cost and durability, low weight, and elasticity [2]. They find their use in diverse applications, such as food packaging, domestic sports and recreations, building and construction, automobile items, electrical devices, farming, healthcare, and plastic furnishings [3]. The demand for plastic products is predicted by the World Economic Forum to increase to 600 million tons by 2025 and may exceed 1 billion tons by 2050 worldwide [4]. Microplastics (MPs) have currently become a global issue because they are released all over the world [5,6]. It was projected that plastic emissions in the aquatic environments of 173 countries will range between 20 and 53 Million tons (Mt)/year by 2030 [7,8]. Table 1 summarizes the trend of rapid escalation of plastic production over the years and shows the alarming, extrapolated statistics.

The source of plastics entering the water ecosystem is through the use of food containers, discarded fishing craft, plastic bags, and plastic drink bottles (water and cold drinks) [9,10]. These are brought to the seas through the action of rivers, floods, and winds, disturbing their gentle balance. MPs are reported to be present at all levels of aquatic environments [10,11] and a large amount of MPs land up in marine environments at an alarming rate. MPs comprise a heterogeneous mixture of varying morphologies, such as fragments, fibers, spheroids, granules, pellets, flakes, or beads in the range of 0.1–5000 µm, and in addition to microplastics, researchers are also concerned with the nanoplastics that are produced from their degradation. Nanoplastics (0.001–0.1 µm) and microplastics (<1.5 µm) are able to translocate across the gut epithelium, causing systemic exposure [12,13,14].

Environmental activists have long highlighted the effect of plastic debris on marine ecosystems and animals; however, it is only recently that the threats posed by microplastics have been highlighted as a grave concern. With the increasing focus on microplastics in the marine environments, a number of studies inspecting the guts of marine invertebrates, such as pelagic fishes, estuarine crustaceans, shrimp, and bivalves have shown the presence of microplastics. Moreover, MPs accumulate in sediments, and the benthic species that thrive in these zones are exposed to them [13,15]. MPs are also reported to affect predatory behavior in fish and cause misunderstandings between MPs and genuine prey (since they are of similar size to planktons) [16]. This way, an injection of MPs will lead to malnutrition as they are stored in key organs, such as the gills, gut, and stomach [17,18,19]. MPs can lead to tissue damage and oxidative stress, and changes in immune-related gene expression can affect the antioxidant status in fish as well. Additionally, fishes are reported to suffer from neurotoxicity and growth and behavioral abnormalities. MPs were also found in fish muscle/meat, which are mainly consumed by humans [20,21,22,23,24]. Growth abnormalities, hormone disruption, metabolic perturbation, oxidative stress, immunological and neurotoxicity malfunction, and genotoxic behavioral alterations have been reported to be caused by an internal build-up of MPs [18,25]. As fish are a major source of protein for humans, MP toxicity may influence aquatic food security [26,27]. The harmful effects of MPs in humans are underexamined. The crucial concern here is that MPs represent a risk to ecosystems and human health and their effects on human health and welfare are entirely undisclosed.

The following review presents an up-to-date comprehensive survey on the impacts of MPs on edible marine and freshwater aquatic species. A regional survey of the studies conducted with respect to marine fishes has been conducted and presented. The comparatively less studied freshwater fishes, shrimps, and crabs have also been reviewed. The ground reality of the current scenario of MP pollution as a threat to human welfare and the possibility of MP-contaminated organisms entering the human food chain has been discussed. Mitigation methods have been briefly touched upon.

## 2. Comprehensive Overview of MP Impacts on Edible Marine and Freshwater Organisms

The overall scheme of microplastic contamination in marine edible fishes, shrimps, crabs, dry fishes, canned fishes, and seaweeds and its rout to humans is given in Figure 1.

### 2.1. Reports of MPs in Marine Fishes

Fishes have always been a major component of human diet and health owing to their high nutrition content, involving essential proteins; polyunsaturated fatty acids, such as omega-3 and omega-6 (which play a key role in preventing atherosclerosis and thrombosis); and vital minerals, such as iron (Fe), calcium (Ca), zinc (Zn), phosphorus (P), selenium (Se), fluorine (F), and iodine (I) [28,29]. Therefore, the consumption of fish from marine or freshwater origins is escalating worldwide. This being the case, MP contamination in fish will eventually adversely affect humans. Thus, MP contamination in edible fish is now one of high concerns, and is under serious scrutiny around the globe. A region-wise consolidation of the various reports on the discovery of MPs in marine fish is presented below.

#### 2.1.1. In South America

A vast array of studies have been conducted on MP ingestion by marine fishes all across the globe. MP ingestion in fishes has been assessed to understand the role of feeding habits of fishes, such as *Centropomus undecimalis*, *Bairdiella ronchus*, and *Gobionellus stomatus*, which dwell in the Estuarine Complex of the Santa Cruz Channel, Brazil [30]. The authors found that *Centropomus undecimalis* was the most MP-polluted fish among the three studied. The MPs extracted from these fishes were in the form of fibers, pellets, and fragments.

In another study [31], the ingestion of microplastics (MPs) by Longnose stingrays in the Western Atlantic Ocean was investigated, and the authors found that 23 specimens of *Hypanus guttatus* from the Brazilian Amazon coast had microplastic particles in their stomach. Fibers were most abundant (82%), blue was the most frequent color (47%), and Polyethylene Terephthalate (PET) was the most frequent polymer recorded (35%).

Other authors investigated the ingestion of microplastics and artificial cellulose particles by 103 specimens of 21 reef fish species from the southwestern Atlantic. The tomtate grunt, *Haemulon aurolineatum*, ingested the most compared with other species. Transparent particles were commonly reported, and polyamide plastic material predominated. Household sewage, fishery activity, and navigation appear to be the principal sources of the artificial particles ingested by the reef fishes [32].

The occurrence and distribution of MPs in the gastrointestinal tracts (GITs) of marine commercial fish species (*Micropogonias furnieri*) from the Bahía Blanca Estuary (BBE) in Argentina was assessed [33,34]. A total of 241 MPs were removed from the GITs of all fish. They were categorized as fibers (60.8%), pellets (28.9%), fragments (8.6%), and laminas (1.4%), and they ranged in size from 0.98 to >5 mm.

The tropical transfer of MPs in different species from Latin American countries has been addressed [35]. The presence of microplastics in seafood, such as *Hoplosternum littorale*, *rachurus murphyi*, *Strangomera bentincki*, *Merluccius gayi*, *Eleginops maclovinus*, *Aplodactylus punctatus*, *Basilichthys austrails* and *M. furnieri* has been reported.

#### 2.1.2. In Europe

MP contamination in the digestive tract of three different wild fishes, namely *Dicentrachus labrax*, *Trachurus trachurus*, and *Scomber colias* from the northeast Atlantic Ocean, and their neurotoxicity and lipid oxidation impacts were studied [21,27]. In all the three species assessed, MPs were found to accumulate in the gastrointestinal tract, gills, and dorsal muscles, causing oxidative damage. Furthermore, it was ascertained that the cause for such oxidative damages was either plastics or MPs-associated chemical compounds. Neves et al. [31] analyzed Portugal’s commercial fish samples (263 fish from 26 commercial species) and discovered MPs in 20% of the fish population from their GI system, of which 67% contained at least one plastic particle [36].

Demersal fishes were not an exception for MPs contamination, with MPs detected in demersal fish species, such as Atlantic cod; *Gadus morhua*, common dab; *Limanda limanda*, and European flounder; *Platichthys flesus*; and two pelagic fish species, such as Atlantic herring (*Clupea harengus*) and Atlantic mackerel (*Scomber scombrus*), which were collected from the North Sea and the Baltic Sea [32]. Out of 290 GI tracts analyzed in the study, 5.5% of the samples contained MPs. Atlantic cod and mackerel had the lowest number of particles in their guts [37,38]; however, 2% of the Atlantic herrings from the North Sea contained MPs [38]. In another study, MPs from four edible fishes, such as Atlantic herring, sprat, common dab, and whiting from the southern North Sea bounded by the coasts of the Netherlands, Belgium, France, and Great Britain, were investigated [39]. Out of the 400 individual fishes investigated, the authors detected two PMMA MPs in spray touts (0.25%, with a 95% confidence interval of 0.09–1.1%).

#### 2.1.3. In Africa

Naidoo et al. [40] assessed the ingestion of MPs in mullet *Mugil cephalus* in Durban Harbour, KwaZulu-Natal, South Africa. Out of the 70 mullet specimens studied, almost all the fishes tested had MPs (on an average 3.8 ± 4.7) in their guts and 73% of these MPs were fibers. The extraction of MPs from the GI of fishes, such as *Trachurus capensis*, *Merluccius capensis*, *Merluccius paradoxus*, *Etrumeus whiteheadi*, *Scomber japonicus*, *Chelidonichthys capensis*, and *Argyrozona argyrozon* from the Agulhas Bank, south of South Africa, was reported. The MPs were of a fibrous morphology, predominantly occurring on an average of 2.8 to 4.6 items/fish [41].

The MP content of two fish species in the Nile River, the Nile tilapia (*Oreochromis niloticus*) and catfish (*Bagrus bayad*), which were sampled from the Nile River in Cairo, Egypt, were checked for MP content in their GIT. The fiber MPs were the most abundant, and more than 75% of the samples contained MPs in their GIT, with PE, PET, and PP as their plastic components [42]. From the Ethiopian lake, Lake Ziway, four commercially important fish species such as *Oreochromis niloticus*, *Clarias gariepinus*, *Cyprinus carpio* and *Carassius carassius*, were examined for the MPs pollution in their GI content [43]. The authors reported that only 35% of fishes sampled were found with ingested MPs, MPs which were predominately fibers with polymer compositions such as PP, PE and alkyd-varnish. These studies indicate that MPs pollution in the Ethiopian freshwater system was lower than that of Alexandria fish and indicate that MPs pollution seems to be a serious threat for fish in River Nile countries.

Adika et al. [44] studied MPs content in fishes such as *Sardinella maderensis*, *Sardinella aurita* and *Dentex angolensi* from the Guinea current region off Ghana’s Coast. From their results, *Sardinella maderensis* were found to contain the highest quantity of MPs (41%) than *Dentex angolensis* (33%) and *Sardinella aurita* (26%) with an average of 40.0 ± 3.8, 32.0 ± 2.7 and 25.7 ± 1.6, respectively with diverse shapes [44]. However, the average quantity of MPs in *Siganus rivulatus*, *Diplodus sargus*, and *Sardinella aurita* from Urban Harbor, Mediterranean Coast of Egypt were 7527, 3593, and 1450 MPs/fish, respectively [45], pointing out to the fact that MPs pollution was very high in Egypt compared to the interior African countries.

#### 2.1.4. In Asia

PVB (poly(vinyl butyral), PVDF (poly(vinylidene fluoride), PtBS (poly(2,4,6,-tribromostyrene), chlorosulfonated-PE, and PVF (poly(vinyl formal)) MPs, including particles and fibers, were detected from the gills and muscles of fishes dwelling in different epipelagic, pelagic demersal/benthopelagic, and coral reef habitats found in seas around Saudi Arabia [46]. Rochman et al. (2015) [47] analyzed MPs from fish samples obtained from commercial fish markets in Indonesia and California, USA. They collected 76 fishes spread over 11 species from a fish market in Indonesia and found that the GI of 28% of fishes contained MPs. The other fish samples (64 individual fish (12 species)) from California were also detected to contain microplastics (25%).

Similarly, MPs from gills, guts, liver, and muscles were analyzed from tissues of commercial species of Asian seabass (*Lateolabrax maculatus*) caught from coastal estuarine areas of China. The study recorded 22–100% and 22–89% MPs from the gills and guts of the tested fishes, respectively; however, there were no MPs found in the liver and muscle except for a few handful of contaminants [48]. Although MPs were detected from fish GITs, the particles were absent in muscles and liver. MPs were also obtained from cultured hybrid fishes from the Pearl River Estuary, South China [49]. All the GIT samples invariably revealed the presence of MPs, with an average abundance of 35.36 n/individual and fiber-shaped (70.1%) particles as predominated ones. In addition to that, MPs content was analyzed from the GIT of 263 commercial fishes (26 species with 63.5% and 36.5% belonging to benthic or pelagic fish communities, respectively) off the Portuguese coast, of which 19.8% of the fish ingested at least one or more MP [36]. The predominant particles were fibrous in nature and were composed of PP, PE, alkyd resin, rayon, PE, nylon, and acrylic polymers.

Similarly, MPs from gills, guts, liver, and muscles were analyzed from Asian seabass (*Lateolabrax maculatus*) tissues obtained from the coastal estuarine regions of China. The study recorded 22–100% and 22–89% MPs from the gills and guts of the tested fishes, respectively; however, there were no MPs found in their liver and muscle except for a few contaminants [48]. Zitouni et al. [50] reported MPs (2.90 ± 1.47 MP/g) contamination in the tissues of the edible commercial fish, *Serranus scriba*, caught from the coastal waters of Tunisia. Furthermore, commercial fishes such as *Sarpa salpa* and *Liza aurata* from the lagoons of Bizerte and Ghar El Melh in Tunisia were detected for MPs in their stomach in Ref. [51]. The fishes from the lagoon of Bizerte (65.33 ± 6.50 and 66.40 ± 5.12 items/individuals in *L. aurata* and *S. salpa*, respectively) contain more MPs than the lagoon-based Ghar El Melh fishes (42.00 ± 6.08 in *S. sapla* and 22.40 ± 3.97 in *L. aurata* items/individual, respectively).

MPs in commercial marine fishes, namely *Atule mate*, *Crenimugil seheli*, *Sardinella fimbriata*, and *Rastrelliger brachysoma*, in the northwest peninsular offshore region of Malaysia were assessed by Foo et al. [52]. Of the 72 commercial fish guts (four species), almost all the fishes contained MPs, and Sardinella fimbriata showed the highest levels of MPs contamination (6.5 ± 4.3 MPs/per fish).

#### 2.1.5. In Australia

There exists a critical knowledge gap on the potential MP transfer to Australian consumers, which is of particular concern as many crustaceans are low-trophic-level scavengers or filter feeders and may be at higher risk of MP contamination [53]. Wooton et al. [54] assessed the abundance of MPs from common commercial fishes, such as goatfish, sea mullet, paddle tail, and common coral trout, from Fiji and Australian fish markets. They reported that the MPs were more prominent in the fishes from Australia (61.6%) than Fiji (35.3%), with an average of 1.58 ± 0.23 pieces/fish and 0.86 ± 0.14 pieces/fish, respectively. Although the morphology of the MPs was diverse, polyolefin was the most abundant polymer type in Australian fish, while film was more common for fish from Fiji. Fang et al. [55] and Ribeiro et al. [56] analyzed muscle tissue from freshly dissected whole point-of-sale fish. Fang et al. [55] did not detect any MPs in either of the analyzed fish species. On the other hand, MPs were successfully detected in *Sardinops neopilchardus* muscle and skin tissues [56]. The authors suggested that the MPs may have adhered to the skin of the fish during commercial packing throughout the supply chain. Overall, three of the four studies analyzing skinless fish muscle tissue did not detect MPs. These findings may suggest that MPs do not translocate across the GIT into the muscle tissues, or that the detection limits for these studies precluded quantification.

#### 2.1.6. In Western America

Microplastic ingestion in marine fish is well documented [57] and authenticated by a number of field studies reporting microplastic ingestion in wild-caught fish of both commercial and non-commercial interest from a broad range of trophic levels, habitats, and benthic zones [33,58,59].

In a previous study, fishes such as gizzard shad and 24 largemouth bass caught from two agricultural reservoirs in the midwestern USA were subjected to MPs detection [60]. Researchers found that these fishes were invariably contaminated (1–49 No/Fish), with MPs in the guts and gills (bass had higher concentrations than shad fishes).

### 2.2. Reports of MPs in Freshwater Fishes

MPs content in natural and farmed freshwater fishes, namely *Prochilodus magdalenae*, *Pimelodus grosskopfii*, and *Oreochromis niloticus* from the Huila region in Colombia were studied by Garcia et al. [61], and their results revealed the presence of fragmented PET, PES, and PE MPs in the stomach, gills, and flesh of both farmed and natural fishes. Munno et al. [62], conducted a survey of MPs ingested in seven different fish species from Lake Ontario and Lake Superior and found that the test fishes (212 fish spanning eight species) from nearshore Lake Ontario, the Humber River, and Lake Superior had 12,442 MPs in their guts, of which PE and PET MPs were found to be predominant. Furthermore, MPs were also assessed from the stomach of the benthic fish *Clarias gariepinus*, which is the top predator of the Vall River, South Africa. The authors reported that each fish contained 7.47 particles/fish; however, the content of MPs in the sediments and water was 46.7 particles/kg and 3300 particles/m^3^, respectively, indicating that the river is heavily loaded with MPs [63].

### 2.3. Reports of MPs in Edible Parts of Fishes

The edible parts (skin and muscles) and non-edible parts (gills and viscera) of the commercial pelagic food-fishes of Indian fishes, such as Indian oil sardine (Sardinella longiceps), Indian mackerel (*Rastrelliger kanagurta*), Malabar thryssa (Thryssa dussumieri), chacunda gizzard shad (*Anodontostoma chacunda*), goldstripe sardinella (*Sardinella gibbose*), Indian anchovy (*Stolephorus indicus*), rainbow sardine (*Dussumieria acuta*), obtuse barracuda (Sphyraena obtusata), and Indian horse mackerel (*Megalaspis cordyla*), caught from Kerala, India, were examined from reminiscent MPs. A total of 163 particles consisting mainly of fragments (58%) were isolated. Out of the fishes tested, 41.1% had MPs in their inedible tissues while only 7% of them had MPs in their edible tissues, with an average of 0.07 ± 0.26 items/fish [64]. Abidin et al. [65] analyzed presence, abundance, and characteristics of MPs in the edible tissues of commercially important pelagic fish species, namely *Parexocoetus mento*, a flying fish from the Indonesian fish market at Bintaro Lombok, West Nusa Tenggara. The study revealed that fragments (368.67 particle/fish) were the most abundant MPs, which were followed by film (263.67), foam (219.33), pellets (173.00), and fibers (62.67). Ribeiro et al. [51] developed a novel method to detect and quantify MPs from tissue samples using a Pyrolysis GC-MS method and applied it to study the MPs in commercial-value Australian seafood, such as oysters, prawns, squid, crabs, and sardines. By their new method, they could detect and quantify PS, PE, PVC, PP, and PMMA in the edible portions of five different seafoods. They found that MPs were invariably found in all the samples tested, in which sardines and squid contained the highest (0.3 mg g^−1^ tissue) and lowest (0.04 mg g^−1^ tissue) total plastic mass concentrations, respectively. A 100 g serving of sardines and squid could potentially have 30 mg and 0.7 mg of MPs, respectively. Akoueson et al. [66] analyzed the MPs from the edible and non-edible parts of commercially important finfish and shellfish, namely Scottish haddock (*Melanogrammus aeglefinus*), Greek seabass (*Dicentrarchus labrax*), Icelandic plaice (*Pleuronectes platessa*), Atlantic mackerel (*Scromber scombrus*), Patagonian scallop (*Zygochlamys patagonica*), and Scottish scallop (*Pecten maximus*). The authors reported high concentrations of MPs in Patagonian scallop, and the particles were identified as PET and PE derivatives. Su et al. [43] reported that there were no MPs found in the muscle and liver tissues of 13 commercially important fishes tested from Hangzhou Bay and the Yangtze Estuary in China. However, MPs were detected in the gut (22–100%) and gills (22–89%) of the individuals tested.

### 2.4. Reports of MPs in Dry Fishes

Karami et al. [67] also studied the presence of MPs in dried fishes in packets (*C. subviridis, J. belangerii, R. kanagurta*, and *S. waitei*) from 14 different dried fishes from seven Asian countries, purchased from local markets in Malaysia. They reported that 246 MPs from these dried-fish sources were of various morphologies, such as fragments, films, filaments, beads, and foams. The highest number of particles (1.92 ± 0.12/individual or 0.56 ± 0.03/gram) were present in *Etrumeus micropus* from Japan. The MP fibers were found to be predominant (~80%), and MPs from the samples tested included PE (35%), PET (26%), PS (18%), PVC (12%), and PP (9%). MPs (51%) isolated from the viscera, gills, and eviscerated flesh of dried fishes, such as greenback mullet (*Chelon subviridis*), Belanger’s croaker (*Johnius belangerii*), Indian mackerel (*Rastrelliger kanagurta*), and spotty-face anchovy (*Stolephorus waitei*). Furthermore, the presence of MPs in 14 marine dried-fish products from seven Asian countries had been assessed by Piyawardhana et al. [68], and most the dried fishes were detected to contain MPs that had a fibrous structure (~80%) and PE, PET, PS, PVC, or PP as their polymer component. In addition to that, MPs in anchovy products bought from the local fishing markets in the western Gulf of Thailand contained 0.47–3.18 fibrous MPs/gram of dried fish [69]. Dried-fish samples of Bombay duck (*Harpadon nehereus*) and ribbon fish (*Trichiurus lepturus*) collected from the Bay of Bengal and sold at Cox’s Bazar and Kuakata were also found to contain MP levels of up to 64% and, as in other dried-fish samples, fibers were the most common type of MPs in these samples, consisting of PE, PS, and PA [70].

### 2.5. Reports of MPs in Canned Fishes

Twenty different brands of commonly available canned sprats and sardines were investigated for MPs contamination [71] and the results exhibited the presence of MPs in only four fish products, with one to three particles per can. Consuming canned fish products generally enables lower particle exposure compared with the atmospheric MP load. MPs with polymeric contents inclusive of PET (36.6%), PS (17.6%), PP (13.5%), PS-PP (10.2%), PS-PET (7.9%), nylon (7.1%), PVC (3.9%), LDPE (3.2%) were detected in canned longtail tuna, yellowfin tuna, and mackerel fish samples obtained Iranian hypermarkets [72]. Moreover, prepared and preserved fish, including canned tuna, was the highest-value imported fishery product with respect to Australia in 2016 [73]. According to the authors, MPs detected in pre-packaged and dried eviscerated fish may have been introduced through processing and packaging, or through translocation from the GIT [67,71].

Hussien et al. [74] analyzed the MPs and toxic elements in canned tuna, salmon, and sardines from seven different brands from Taif markets, Kingdom of Saudi Arabia. Except salmon, canned tuna and sardines were found to be contaminated with MPs in edible tissues. Tuna was contaminated with nylon, 1,2-polybutadiene, and ethylene vinyl alcohol while sardines were contaminated with ethylene vinyl alcohol and poly(vinyl stearate). The samples were also reported to be contaminated with Al > Se > Zn and traces of As and Sb. More recently, Diaz-Basantes et al. (2022) detected 692 ± 120 MPs/100 g and 442 ± 84 MPs/100 g in tuna-storing liquids such as water and oil, respectively, from canned tuna samples from Ecuador markets. PET, polystyrene, and nylon were reported to be the most frequently found MPs in canned tuna.

### 2.6. Reports of MPs in Marine Shrimps

Marine organisms continuously interact with MPs in the environment; therefore, MPs are detected more prevalently in crustaceans [75,76]. Shrimp is the most preferable crustacean seafood worldwide and they are supplied through catching from natural habitats, such as fresh, ocean, and brackish waters. On the other hand, a major part of the shrimp supply is from aquaculture [77]. Shrimps, either in the wild or in aquaculture, are constantly being exposed to MPs pollution. Devriese et al. [78] studied the content of MPs in brown shrimp (Crangon crangon) from the shallow coastal water habitats of the southern North Sea and Channel area. Their results confirmed the presence of plastic fibers ranging between 200 and 1000 µm in size in 63% of the shrimps tested.

Brown shrimp (*Metapenaeus monocerous*) and tiger shrimp (*Penaeus monodon*) from the offshore waters of the Northern Bay of Bengal and Bangladesh were also subjected to MP analysis [79], which revealed 3.40 ± 1.23 and 3.87 ± 1.05 MPs/g tissue of *P. monodon* and *M. monocerous*, respectively, and the polymer compositions were identified to be polyamide-6 and six particles of rayon polymers. Gray and Weinstein [80] studied the effect of the shape and size of MPs on the ingestion of adult daggerblade grass shrimp (*Palaemonetes pugio*). The authors observed that the sizes and shapes of MPs play an important role in this case. Eleven different-sized and -shaped MPs (spheres, fragments, and fibers) were tested at a concentration of 5000 particles/L for 3 h. The presence of MPs from shrimp paste purchased from five provinces of the Andaman Sea and the Gulf of Thailand were assessed by the authors in Ref. [81]. They found that the shrimp paste contained 6 to 11.3 MP particles/10 g, with the predominant particles being fibers and fragments (0.1 to 1.0 mm) with PTE, PU, PS, PVC, and rayon polymer compositions.

Abbasi et al. [23] studied the MP content in commercially important fish, as well as a crustacean, *Penaeus semisulcatus*, in the Musa Estuary and Persian Gulf. A total of 828 MPs were obtained from demersal and pelagic fish (*Platycephalus indicus*, *Saurida tumbil*, *Sillago sihama,* and *Cynoglossus abbreviatus*) and in the exoskeleton and muscle of the tiger prawn, *P. semisulcatus*. Of all the species studied, MPs were found to be comparatively lower in *P. semisulcatus* (mean = 7.8) compared with the other commercial fishes tested, and most of the particles were fibrous fragments of diverse colors and sizes. However, another study conducted in the Persian Gulf by the authors in Ref. [82] found that the MPs content was higher in *P. semisulcatus*, with the highest (mean 0.360 items/g muscle) contents recorded in this case compared with other fishes. These findings contradicts those of Ref. [23].

In another study, MPs in the stomach of an economically and ecologically key species, *Aristeus antennatus*, of the Mediterranean deep-sea was studied. MPs were present in 39.2% of the individuals; however, the samples collected near Barcelona invariably showed the presence of MPs ingestion (100%). Most of the particles were fibers; however, they exerted no negative effects on the shrimp’s biological conditions [83]. Ingested MPs were also detected in two deep-sea crustaceans, namely *Nephrops norvegicus* and *Aristeus antennatus*, and the study exhibited the presence of MPs in almost 60–85% of the specimens from the studied sites.

The presence of MPs in two economically and ecologically key crustacean species, namely the Norwegian lobster Nephrops norvegicus and the shrimp *Aristeus antennatus* were found in 14 sites around Sardinia Island in Mediterranean Sea [84]. Out of the samples tested, 83% with 5.5 ± 0.8 MPs/individual and 67% with 1.66 ± 0.1 MPs/individual of *N. norvegicus* and *A. antennatus*, respectively, were reported. Since those organisms that contain MPs are deep-sea dwellers (sampled between depths of 270 and 660 m), these can be considered as valuable bioindicators for deep-sea MP pollution. Furthermore, Curren et al. [85] studied MP pollution in *Litopenaeus vannamei* (Pacific white leg shrimp), *Pleoticus muelleri* (Argentine red shrimp), and *Fenneropenaeus indicus* (Indian white shrimp), which are commonly available shrimps in Singaporean supermarkets. It is quite evident from this study that all the species tested were polluted with film- and sphere-shaped MPs. *L. vannamei* (93–97%) individuals were the most likely to contain film-shaped morphologies of MPs and *P. muelleri* (70%) and *F. indicus* (61%) primarily contained sphere-shaped morphologies.

Seasonal variations in the occurrence of MPs in the marine commercial shrimp species, *Fenneropenaeus indicus*, in the coastal waters of Cochin, India, were studied for a period of 12 months, from March 2018 to February 2019. Fibrous MPs (83%) were the most abundant of the 128 MPs detected from the soft tissues of 330 shrimps and each shrimp contained an average of 0.39 ± 0.6 MPs/wet weight. In another study conducted by Gurjar et al. [86], shrimps such as *Metapenaeus monoceros, Parapeneopsis stylifera,* and *Penaeus indicus* from the northern part of the Arabian Sea in India had MPs in their gastrointestinal tracts. The study revealed the presence of 6.78 ± 2.80 items per individual, with different colors and shapes, such as fibers, fragments, pellets, beads, and films. These studies indicated high levels of MP pollution in the west coast of India [64,86]. MPs were also studied in brown shrimp (*Metapenaeus monocerous*) and tiger shrimp (*Penaeus monodon*) from the Northern Bay of Bengal in Bangladesh. The GIT of *P. monodon* and *M. monocerous* contained 3.40 ± 1.23 and 3.87 ± 1.05 MPs/g GT, respectively [79], among which MPs with polyamide-6 and rayon polymers predominated.

### 2.7. Reports of MPs in Freshwater Shrimps

Freshwater prawns are also a crustacean delicacy; therefore, MP particles in freshwater prawn species are becoming a global concern. However, regarding the MP pollution in inland freshwater prawns, comparatively very few reports are available compared with marine shrimps. Nan et al. [87] studied the MP pollution in *Paratya australiensis*, an Australian freshwater shrimp. From this study, they found that the prawn samples contained 0.52 ± 0.55 MPs/individual, in which the composition of the dominant polymer (MPs) in the shrimps was identified to be rayon, whereas in water samples, the prevalent particles were made of polyester [87]. The gastrointestinal tracts of two economically important giant freshwater prawns, namely *Macrobrachium rosenbergii* and the white leg shrimp *Litopenaeus vannamei*, were studied from subjects obtained from a polyculture pond in Thailand. An analysis for the presence of MPs showed that each individual organism contained about 33.31 ± 19.42 (*Macrobrachium rosenbergii* male), 33.43 ± 19.07 (*Macrobrachium rosenbergii* female), and 11.00 ± 4.60 (white legged prawns) MP particles per individual [88]. The authors observed the MPs in the samples to have morphologies that included fibers, fragments, films, and spheres in which fibers were the most predominant and the polymer composition of MPs was detected as polyethylene, polycaprolactone, polyvinyl alcohol, and acrylonitrile butadiene styrene.

### 2.8. Reports of MPs in Crabs

Most of the crab species feed on the organic debris sedimented on the ocean floor; therefore, MPs sedimented along with the organic debris expose the bottom dwellers, especially crabs, to a higher risk of obtaining MPs and their associated toxicants. The extent of MPs contamination was assessed in the gills and digestive tract of sand crabs or Pacific mole crabs (*Emerita analoga*) collected from Del Monte Beach in Monterey Bay, CA [89]. Out of the 18 crabs collected, 12 crabs contained an average number of 5 particles. Kleawkla et al. also detected MPs from blue swimming crabs, *Portunus pelagicus*, from Wonnapha coastal wetland, located along the Gulf of Thailand in Chonburi Province. The author had examined a total of 296 crabs. A total of 26.35% of the samples had MPs with an average of 0.73 ± 1.4 items/crabs, in which plastic debris and fibers were the dominant MPs. Watts et al. (2014) reported the uptake mechanism of polystyrene MPs using fluorescently labeled particles and found that it retains for almost 14 days in the body tissues of the crabs. This is very important information since it can cause public awareness. Accumulation of MPs in tissues of four different species of wild crabs (*Portunus trituberculatus*, *Charybdis japonica*, *Dorippe japonica*, and *Matuta planipes*) from the fishing grounds of Haizhou Bay, Lvsi, and the Yangtze River Estuary in China were studied for the abundance and composition characteristics of MPs [90]. The study revealed that 89.34% of the samples contained MPs, with an average of 2.00 to 9.81 items/individual and 0.80 ± 1.09 to 22.71 ± 24.56 items/g wet weight. It was noted that although the gills and guts had abundant MPs, the muscles had no MPs.

Goswami et al. [91] traced the MPs in the blue swimmer crab, *Portunus pelagicus*, which was sampled from Port Blair Bay and the Andaman and Nicobar Islands in the Indian Ocean, along with other species, such as zooplankton, finfish, and shellfish. Their study revealed that 80% of the samples analyzed showed the presence of MPs with fragments and pellet structures. Thames Estuary resident brachyuran crab species, such as *Carcinus maenas* (native shore crab) and *Eriocheir sinensis* (the invasive Chinese mitten crab) were analyzed for the MPs in their gills and gastric mill, and based on their results, it was evident that E. sinensis was invariably contaminated with MPs, whereas 71.3% of the *C. maenas* sampled showed MPs in the form of fiber, film, fragments or tangled fibers [92]. Yet, another study was conducted to assess MP pollution in blue crabs (*Callinectes sapidus*) sampled from 12 sampling sites in Corpus Christi Bay, Texas. From the 39 blue crabs analyzed, 36% of the collected blue crab samples contained fully synthetic fragments within their stomachs, with an average of 0.87 items per crab [93]. The prevalent dwellers of the Californian coast, the Pacific mole crabs (*Emerita analoga*) were collected from 38 beaches, and it was found that the crabs (35%) were detected with ingested MPs that were abundantly available in the sediments of California beaches [94]. Ingested MPs in spider crabs (*Maja squinado*) from the Celtic Sea were assessed along with other bottom dwellers, such as *Pleuronectes platessa* and sand eels (*A. tobianus*). From the study, the authors reported that all three studied species were contaminated with MPs to various extents (*P. platessa* (50%), *M. squinado* (42.4%), and *A. tobianus* (44.4%) [95]. For the first time, MP-contaminated crabs and other bottom dwellers of the Arctic Chukchi Sea were screened by Fang et al. [96]. The study was conducted with a total of 413 dominant benthic organisms from 11 different species dwelling on the shelf of the Bering and Chukchi Seas, revealing that the benthic samples from all sites contained MPs that ranged from 0.04 to 1.67 items/individual. Of these, the crab *P. borealis* was classified under organisms that were reported with high quantities of MPs, which were generally fibers (87%) and films (13%) of PA, PE, PET, and cellophane (CP).

### 2.9. Reports of MPs in Seaweed

Many seaweeds are edible and are part of the diets of many Asian countries and some European nations. According to the Food and Agriculture Organization of the United Nations, aquatic plant production grew from 13.5 million tons to over 30 million tons from 1995 to 2016 [48,97]. China is the world’s largest producer of seaweed, with a long-term mariculture of nori seaweed, *Pyropia yezoensis* [98]. Seaweeds are highly nutritious since they contain huge quantities of omega-3 fatty acids and essential minerals, such as iodine, iron, copper, zinc, selenium, and bioactive compounds, which are very essential for many vital human metabolic activities [99]. On the other hand, in the ocean, seaweed serves as a habitat for a variety of bio entities since it provides food and shelter for those organisms [100], and with their enhanced pollutant bioaccumulation potential, they are also being used as ocean pollutant monitors [101]. Seaweeds are the prime source of microplastic pollution transfer to many trophic levels; thus, they can be valuable sources for the identification of MPs.

Fucus vesiculosus, a medicinal seaweed with high iodine and mineral contents, was used by Lars Gutow et al. [102] to study the adherence of MPs to its surface, in addition to studying further trophic transfer (as vectors) using the periwinkles snail model under laboratory conditions. They found that the MPs were in the stomach and gut of the snail because as soon as the snail fed on seaweeds after digestion, most of the MPs were released along with fecal material. Yet, another study by Anirut Klomjit et al. [103] revealed the presence of MPs on the surface of an edible red seaweed Gracilaria fisheri and a green seaweed *Caulerpa lentillifera* from aquaculture. This was quantified and the study exhibited the presence of particles ranging from 16.46 ± 2.56 to 181.73 ± 86.42/100 g wet wt in the two seaweeds studied, where the highest particle count (181.73 ± 86.42 MPs/100 g wet wt) was found on *G. fisheri*. Sundbæk et al. [104] studied the adherence of commercially available fluorescent polystyrene microplastics (PS MPs) on the edible seaweed *Fucus vesiculosus*, which was investigated in the laboratory, and the results revealed that the most significant part of the PS MPs adhered to the surface of the tested seaweed.

Prihandari et al. [105] optimized a method for digesting MPs to analyze MPs in dried *Gracilaria fisheri* seaweed. The authors employed digestive methods, such as cellulase-based enzymatic and H_2_O_2_-based oxidative methods, and a combination of enzymatic and oxidative digestion methods for the analysis of MPs from the dried samples of edible red seaweed, *Gracilaria fisheri*. Among the methods tested, the authors claimed that the cellulase-based enzymatic method had moderate digestion efficiency (59.3–63.7%), with a high polymer recovery rate (94.7–98.9%).

Furthermore, the oxidative digestion method using fishes that are generally consumed whole, such as the sardines, anchovies, and sprats of the Spanish Mediterranean coast, were also studied for their MP contents, and the results showed that 14–15% of the tested samples had MPs or natural fibers in their GI tract. In addition to that, the MPs content in Atlantic herring and European sprat have been investigated. The study was conducted on fish samples caught over a period of 30 years in the Baltic Sea, and they found that around 20% of the fish contained MPs in their GI tract. Yet, another MPs assessment in the Baltic Sea showed the presence of only 1–2% of MPs in Atlantic herrings and sprats. O_2_ had a high digestion efficiency (93.0–96.3%) and polymer recovery rate (>98%). Finally, the combination method was claimed to be the best method for seaweed digestion for MPs analysis. All those digestion methods showed no impact (chemical changes) on the spiked PE, PP, PS, PVC, and PET polymers after the digestion process.

Seng et al. [106] quantified the density of MPs on the surfaces of intertidal seagrasses, i.e., *Cymodocea rotundata, Cymodocea serrulate*, and *Thalassia hemprichii*, and two subtidal macroalgal species, such as *Padina* sp. And *Sargassum ilicifolium*, and their study exhibited that the MPs densities were significantly higher on seagrasses than on macroalgae and there was no relationship between microplastic density and epibiont cover in either seagrass or macroalgae. The MPs detection method from seaweed was optimized using Nile red stain [103]. Microwave-assisted nitric acid digestion was found to be the optimum method for the detection of MPs from seaweed when PMMA, PS, PE, PA, PVDC, UHMW PE, LDPE, PET, PHB/PHV(2%), PP, and PVC (almost globular shaped particles) were used.

Although the presence of MPs was detected on the surfaces of the whole thalli of the edible seaweed nori (*Pyropia* spp., *Ulva prolifera*, *Sargassum horneri*, *Cladophora* sp., *Undaria pinnatifida*, *Ulva pertusa*), during the culture period, the edible seaweed *Pyropia* spp. contain higher abundances of MPs (0.17 ± 0.08 particles/g fresh wt) than other macroalgae (0.12 ± 0.0 9 particles/g fresh wt) [20,98,107]. According to Li et al. [107], nori seaweed was found to attract the highest quantity of MPs (1.53 ± 0.72 items/gr dry wt) compared with the other seaweeds that occur naturally. Li et al. [107] and Feng et al. [108] detected 11 different polymer compositions and PE, PET, PP, PS, rayon, and cellophane (CP) were the main ones. Table 2 gives an overview of the region-wise reports on MPs detected in marine fishes.

## 3. Future Perspectives and Mitigation Measures

Figure 2 represents the effects of various MPs on seafood and related organisms linked in the food web. The effects on marine organisms such as fish, crabs, shrimps, and seaweeds have been well reported; however, the effects of MPs as they travel through secondary consumers (of MPs) is unreported and unexplored, making it an area for future research. When it comes to aquatic organisms, as deduced through the course of the review, deleterious effects have been observed. Organisms that swallow plastic specks of no nutritional value have been reported to have struggled to consume food, indirectly leading to death. Investigators autopsied sea turtles that they found dead on beaches and they found plastics in their guts and chemicals in their tissues. In 2020, the same team completed a set of analyses for nine hawksbill turtle hatchlings, under 3 weeks old. One hatchling, which was only 9 cm long, had 42 plastic pieces (mostly microplastics) in its gastrointestinal tract [115]. Apart from the toxic effects on the fishes, the more dreaded threat is when through the polluted fishes, these microplastics enter the human food chain. Microplastics are extremely complex and what is most dreaded is that these could have a deleterious impact on humans. MPs have been found in edible fish according to various research, and as a result of biomagnifications, MPs penetrate human systems [116,117,118]. Although the risk of plastics to humans is not yet established, their occurrence in food and water destined for human consumption has been reported. The prevalence of micro-sized plastics in the ecosystem and living organisms and their trophic transfer along the food web has become subtly evident. Especially with respect to fish and aquatic organisms, although massive amounts of research have been generated in this aspect, a collation of available data with a pertinent hazard evaluation remains difficult. This is because the story does not end with the fish—it moves on into other aquatic predators, birds, animals, and humans. Therefore, the microplastics are lost in this intense web and an actual percentage of their threat is still unassessed.

Table 3 lists recommendations on future directions for research. The risk of ingestion of microplastics by humans through eating contaminated seafood has been proven to be low, because in most cases the gastrointestinal (GI) tract of the fishes, where the microplastics predominantly accumulate, is discarded. The actual risk comes from mussels, oysters, and animals eaten whole or with their GI tract still intact, such as in the case of shrimps and crabs. Therefore, although there is a specified quantity reported in fishes and other edible aquatic organisms, only a fraction of these MPs will progress to human food. An EFSA panel report [119] suggests that following oral ingestion, the largest fraction (>90%) of the ingested micro- and nanoplastics will be excreted via feces, and particles smaller than 150 µm may cause systemic exposure. Since this is the size range that can translocate across the gut epithelium, this is a grey area that is unexplored thus far and deserves systematic assessment. The real concern could be because of fishes that are generally consumed whole, such as the sardines, anchovies, and sprats of the Spanish Mediterranean coast [120]. These are fishes that are consumed whole, and it is known that 14–15% of these fishes had MPs or natural fibers in their GI tract. In addition to that, Beer et al. (2018) [121] also investigated the MPs content in Atlantic herring and European sprat. The study was conducted on fish samples caught over a period of 30 years in the Baltic Sea and they found that around 20% of the fish contained MPs in their GI tract. Yet, another MPs assessment in Baltic Sea validated the presence of only 1–2% of MPs in Atlantic herrings and sprats [122]. Whatever the range of percentages, the MPs in edible fish tissues and those consumed whole have the potential to enter the human system, which means they require serious scrutiny. Crabs and shrimps consumed without the entrails removed, or without being deshelled, are also a means through which MPs could enter the human body; however, this can be averted by avoiding the consumption of whole fish/shrimp/crabs and ensuring that the gut is removed (and possibly also the gills).

A study [123] at Trinity College Dublin proved that kettles and baby bottles also shed microplastics. They reported that when parents prepare baby formula in hot water inside a plastic feeding bottle, their infant might end up swallowing more than one million microplastic particles each day, as per their research calculation. The scale of the issue is massive; however, what are the main culprits and causes, and could there be a solution in sight for this global problem? It is something that needs to be addressed and we must clearly and systematically evaluate this scenario.

In the wild, it is hard to trace and assess the fate of MPs; however, in vitro, laboratory-based pilot studies that mimic a niche can lead to a systematic accumulation and derivation of the kinetics of MPs in an environment. Interestingly, most research is popularly backed up with numerous in vitro/laboratory assessments and in vivo/real-world assessments are lesser in magnitude comparatively. With respect to MPs, we see the vice versa trend; therefore, this review encourages future well-planned pilot studies with set ups that closely mimic real-time systems involving subjects that are distinctly connected in the food web. There is also the possibility that MPs in a fish’s gut may not scale up into human bodies. Therefore, the concern then will be how it may diffuse through the birds and animals that feed on these fishes. A clear understanding will help us to narrow down the affected population. Future perspectives and directions for MPs research should take this direction to yield data to fill this gap.

There is one more gray area that has not been looked into, namely the effect of the additives/plasticizers/dyes contained in MPs on fishes, shrimps, and crabs. As explained earlier, the actual containment of MPs could be limited to the entrails of fishes, which are generally discarded; however, that does not stop the additives, plasticizers, and dyes from diffusing into the edible portions of the fish. It is reported that additives account for around 4% of the total weight of plastics produced [124,125]. Sometimes, additives make up half of the total material [126]. Microplastics in oceans and coastal regions, and those deposited on beaches have been found to contain these additives: polybrominated diphenyl ethers (PBDEs) from 0.03 nanograms per gram (ng/g) to 50 ng/g, bisphenol A (BPA) from 5 ng/g to 200 ng/g, nonylphenol (NP) from 20 ng/g to 2500 ng/g, and octylphenol (OP) from 0.3 ng/g to 50 ng/g [127]. Data on how these additives/plasticizers/dyes behave once they enter the fishes are only sparingly available. Additionally, data on their diffusion into the edible portions of fish are even more sparse. These data are crucial to determining the actual impact of MPs on seafoods. This review draws the attention of the scientific community to look into this neglected aspect of MPs.

Microplastics research is not lacking in numbers of publications; however, it is lacking in focus and objective and subjective solutions. ‘We found microplastics here and we found microplastics there’ are now common headlines, and we must find answers to questions such as how it got in there? What does it do? Where does it go from there? How do we avoid it? To do so, we must use a systematic end-to-end approach. This is the need of the hour. Of course, after all is said, it is mandatory that the use of plastics should be reduced, plastics should be recycled, single-use plastics should be restricted, biodegradable plastics should be developed, and better disposal methods for disposing plastic wastes should be devised. Cutting off the source is the most ideal and fool-proof remedy. Letting things loose into the environment and then trying to gather them up is a wild-goose chase and, as always, prevention is better than cure.

## 4. Conclusions

The current investigations involving marine and freshwater fish, shrimps, and crabs have been consolidated and an executive summary is presented herewith. Whether MPs pose a serious impact on marine fauna and secondary consumers (humans) is difficult to conclude with the available information. Numerous reports exist, yet a systematic analysis of how MPs flow through the food chain and how aggressive they can be remains to be performed. This has been identified as a research gap that needs to be filled, which is a motivation for future studies. Mitigation measures and areas needing attention have also been highlighted.

## Figures and Tables

**Figure 1 foods-11-03976-f001:**
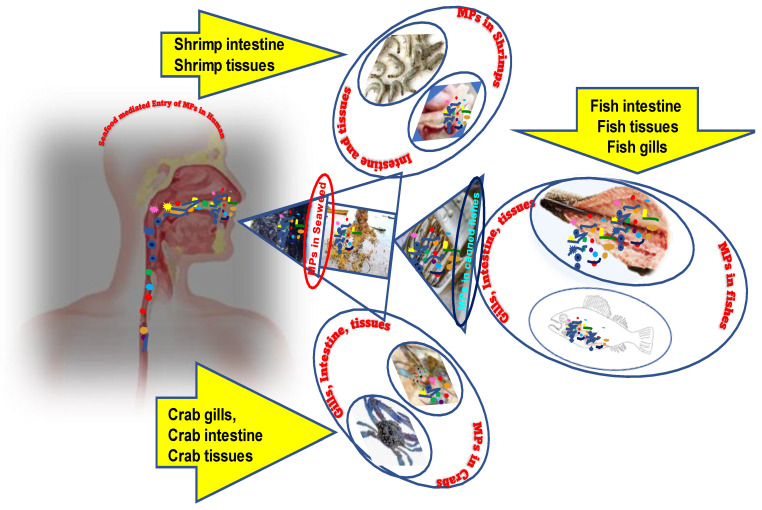
An overview of the various aquatic foods that are contaminated with MPs that have been discussed in the review. The pointers specify the actual locations where MPs are reported to be accumulated in fish, shrimp, and crabs, which enter the human body through MP-contaminated aquatic foods.

**Figure 2 foods-11-03976-f002:**
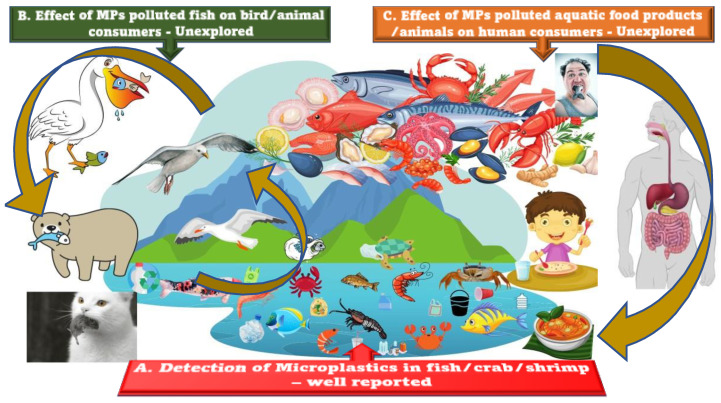
Schematic representation of the major discussion points of this review, which highlights the fact that (A) the detection of MPs in fish/crab/shrimps are supported by extensive reports, while the interlinked aspects of (B) effects of MPs-polluted fish on bird/animal consumers and (C) on human consumers has meagre scientific evidence.

**Table 1 foods-11-03976-t001:** Reports with timelines showing rapid increases in plastic production.

Year	Plastic Production (Million tons)/Year	Reference
1960	0.5	PlasticsEurope, 2018
1917	348	PlasticsEurope, 2018
2018	359	PlasticsEurope, 2019
2018 (China alone)	107.7	PlasticsEurope 2019
2025 (projection)	600	FAO Fisheries and Aquaculture Technical Paper 615, 2017
2050 (projection)	1000	FAO Fisheries and Aquaculture Technical Paper 615, 2017

**Table 2 foods-11-03976-t002:** Region-wise detection of microplastics in marine fishes.

Region/Country	Species	Source	Nature of Processing	Microplastic Shape	Microplastic Quantity	Rate of Occurrence	References
South America/Argentina	*Micropogonias furnieri*	Gastrointestinal tract	Fresh	Fibers (60.8%), fragments (8.6%), laminas (1.4%), and pellets (28.9%)	12.1 ± 6.2 MPs fish^−1^	-	[33]
South America/Brazil	*Bairdiella ronchus*	Stomach and intestine	Fresh	Fibers (62%), fragments (15%), and pellets (23%)	1.2 ± 1.3 MPs fish^−1^	67%	[30]
	*Centropomus undecimalis*			Fibers (28%), fragments (4%), and pellets (68%)	3.3 ± 2.9 MPs fish^−1^	77%	
	*Gobionellus stomatus*			Fibers (71%), fragments (25%), and pellets (4%)	1.7 ± 1.5 MPs fish^−1^	74%	
Europe/Italy	*Boops boops*	Gastrointestinal tract	Fresh	Fiber (87%), fragment (7%), film (5%), and granule (1%)	1.8 ± 0.2 MPs individuals^−1^	56%	109.
Europe/Portugal	*Alosa fallax*	Stomach	Fresh	Fragment	1.0 MPs individuals^−1^	100%	[36]
	*Argyrosomus regius*			Fragment and fiber	4.0 MPs individuals^−1^	60%	
	*Boops boops*			Fragment and fiber	3.0 MPs individuals^−1^	9%	
	*Brama brama*			Fiber	2.0 MPs individuals^−1^	33%	
	*Dentex macrophthalmus*			Fiber	1.0 MPs individuals^−1^	100%	
	*Helicolenus dactylopterus*			-	0.0 MPs individuals^−1^	0%	
	*Lepidorhombus boscii*			-	0.0 MPs individuals^−1^	50%	
	*Lepidorhombus whiffiagonis*			-	0.0 MPs individuals^−1^	0%	
	*Lophius piscatorius*			Fiber	1.0 MPs individuals^−1^	50%	
	*Merluccius merluccius*			Fiber	2.0 MPs individuals^−1^	29%	
	*Merluccius merluccius* (local market)			Fiber	2.0 MPs individuals^−1^	20%	
	*Mullus surmuletus*			Fiber	2.0 MPs individuals^−1^	100%	
	*Mullus surmuletus* (local market)			Fiber	5.0 MPs individuals^−1^	100%	
	*Pagellus acarne*			Fiber	1.0 MPs individuals^−1^	100%	
	*Polyprion americanus*			-	0.0 MPs individuals^−1^	0%	
	*Raja asterias*			Fiber	4.0 MPs individuals^−1^	43%	
	*Sardina pilchardus*			-	0.0 MPs individuals^−1^	0%	
	*Scomber japonicus*			Fragment and fiber	20.0 MPs individuals^−1^	31%	
	*Scomber scombrus*			Fragment and fiber	6.0 MPs individuals^−1^	31%	
	*Scyliorhinus canicula*			Fragment and fiber	3.0 MPs individuals^−1^	12%	
	*Scyliorhinus canicular* (local market)			Fiber	3.0 MPs individuals^−1^	67%	
	*Solea solea*			-	0.0 MPs individuals^−1^	0%	
	*Torpedo torpedo*			-	0.0 MPs individuals^−1^	0%	
	*Trachurus picturatus*			Fiber	1.0 MPs individuals^−1^	3%	
	*Trachurus trachurus*			Fragment and fiber	3.0 MPs individuals^−1^	7%	
	*Trichiurus lepturus*			-	0.0 MPs individuals^−1^	0%	
	*Trigla lyra*			Fragment and fiber	8.0 MPs individuals^−1^	19%	
	*Trisopterus luscus*			-	0.0 MPs individuals^−1^	0%	
	*Zeus faber*			Fiber	1.0 MPs individuals^−1^	100%	
Europe/Italy	*Boops boops*	Gastrointestinal tract	Fresh	Filament (81.95%), film (3.19%), fragment (14.29%), and sphere (0.58%)	2.42 MPs individuals^−1^	47.03%	[109]
Europe/France	*Boops boops*	Gastrointestinal tract	Fresh		1.77 MPs individuals^−1^	47.0%	
Europe/Greece	*Boops boops*	Gastrointestinal tract	Fresh		1.29 MPs individuals^−1^	25.2%	
Europe/Spain	*Boops boops*	Gastrointestinal tract	Fresh		1.85 MPs individuals^−1^	49.7%	
Europe/Spain	*Boops boops*	Gastrointestinal tract	Fresh	Filament	3.75 ± 0.25 MPs individuals^−1^	57.8%	[110]
Europe/Spain	*Boops boops*	Gastrointestinal tract	Fresh	Fragment, fiber, and granule	1.79 MPs individuals^−1^	46.1%	[61]
Europe (North Sea)	*Clupea harengus*	Esophagus, stomach, and intestines	Fresh	-	1.7 MPs individuals^−1^	3.5%	[38].
	*Eutrigla gurnardus*				0.0 MPs individuals^−1^	0.0%	
	*Gadus morhua*				1.1 MPs individuals^−1^	27.9%	
	*Melanogrammus aeglefinus*				1.0 MPs individuals^−1^	12.4%	
	*Merlangius* *merlangus*				1.3 MPs individuals^−1^	11.4%	
	*Scomber* *scombrus*				0.0 MPs individuals^−1^	0.0%	
	*Trachurus trachurus*				1.0 MPs individuals^−1^	2.0%	
Europe (North Sea)	*Clupea harengus*	Stomach and gut	Fresh	-	0.0 MPs individuals^−1^	0.0%	[39]
	*Limanda limanda*			-	0.0 MPs individuals^−1^	0.0%	
	*Merlangius* *merlangus*			-	0.0 MPs individuals^−1^	0.0%	
	*Sprattus* *sprattus*			Spherical	2.0 MPs individuals^−1^	0.7%	
Europe (North Sea and Baltic Sea)	*Clupea harengus*	Gastrointestinal tract	Fresh	Film, fragment, fiber, and spherule	0.0 MPs individuals^−1^	0.0%	[37]
	*Gadus morhua*				1.0 MPs individuals^−1^	1.2%	
	*Limanda limanda*				4.0 MPs individuals^−1^	4.5%	
	*Platichthys flesus*				2.0 MPs individuals^−1^	5.6%	
	*Scomber scombrus*				9.0 MPs individuals^−1^	17.7%	
Africa/South Africa	*Argyrozona argyrozona*	Gastrointestinal tract	Fresh	Fiber (95.14%) and fragment (4.86%)	2.8 ± 0.7 MPs fish^−1^	-	[41]
	*Chelidonichthys capensis*				3.4 ± 0.4 MPs fish^−1^		
	*Etrumeus whiteheadi*				3.3 ± 0.5 MPs fish^−1^		
	*Merluccius capensis*				4.2 ± 0.6 MPs fish^−1^		
	*Merluccius paradoxus*				3.8 ± 0.7 MPs fish^−1^		
	*Scomber japonicus*				4.6 ± 0.8 MPs fish^−1^		
	*Trachurus capensis*				3.9 ± 1.0 MPs fish^−1^		
Africa/South Africa	*Ambassis dussumieri*	Whole fish	Fresh	Fiber (68%), fragment (21%), f. bundle (9%), and film (2%)	0.93 ± 0.75 MPs fish^−1^	69%	[40]
	*Mugil* sp.				1.00 ± 1.46 MPs fish^−1^	55%	
	*Oreochromis mossambicus*				0.59 ± 0.73 MPs fish^−1^	45%	
	*Terapon jarbua*				0.66 ± 0.81 MPs fish^−1^	48%	
Africa/Egypt	*Bagrus bajad*	Gastrointestinal tract	Fresh	Fiber (61.7%), film (29.8%), and fragment (8.5%).	4.7 ± 1.7 MPs individuals^−1^	78.6%	[42]
	*Oreochromis niloticus*			Fiber (65.3%), film (25.6%), and fragment (8.5%)	7.5 ± 4.9 MPs individuals^−1^	75.9%	
Africa/Egypt	*Atherina boyeri*	Gastrointestinal tract	Fresh	Filament, fragment, and glossy sheet	28 ± 21 MPs fish^−1^	-	[45]
	*Boops boops*				213 ± 198 MPs fish^−1^		
	*Diplodus sargus*				3593 ± 3985 MPs fish^−1^		
	*Lithognathus mormyru*				406 ± 484 MPs fish^−1^		
	*Sardinella aurita*				1450 ± 3207 MPs fish^−1^		
	*Siganus rivulatus*				7527 ± 9551 MPs fish^−1^		
	*Sphyraena viridensis*				46 ± 13 MPs fish^−1^		
	*Terapon puta*				122 ± 108 MPs fish^−1^		
Africa/Ghana	*Dentex* *angolensis*	Gastrointestinal tract	Fresh	Fiber, film, and fragment	32.0 ± 2.7 MPs individuals^−1^	-	[44]
	*Sardinella aurita*				26.0 ± 1.6 MPs individuals^−1^		
	*Sardinella maderensis*				40.0 ± 3.8 MPs individuals^−1^		
Africa/Tunisia	*Serranus scriba*	Gastrointestinal tract	Fresh	Fragment	6.11 ± 0.48 MPs g/of tissue	100%	[50]
		Muscle	Fresh	Fragment	6.03 ± 0.47 MPs g/of tissue	100%	
Africa/Tunisia	*Liza aurata*	Gastrointestinal tract	Fresh	Fiber, fragment, and film	43.9 MPs individuals^−1^	100%	[51]
	*Sarpa salpa*				54.2 MPs individuals^−1^	100%	
Asia/Bangladesh	*Anodontostoma chacunda*	Gastrointestinal tract	Fresh	Fibers (53.4%), films (40.0%), fragments (3.3%), foams (1.9%), and granules (1.4%)	1.4 MPs fish^−1^	-	[111]
	*Carangoides chrysophrys*				2.0 MPs fish^−1^		
	*Coilia neglecta*				1.5 MPs fish^−1^		
	*Harpadon nehereus*				1.8 MPs fish^−1^		
	*Megalaspis cordyla*				1.0 MPs fish^−1^		
	*Otolithoides pama*				1.8 MPs fish^−1^		
	*Priacanthus hamrur*				3.8 MPs fish^−1^		
	*Sardinella brachysoma*				2.0 MPs fish^−1^		
	*Sciades sona*				3.0 MPs fish^−1^		
	*Setipinna tenuifilis*				3.2 MPs fish^−1^		
Asia/China	*Konosirus punctatus*	Gastrointestinal tract	Fresh		4.4 MPs species^−1^		[112]
	*Mugil cephalus*				5.2 MPs species^−1^		
Asia/China	*Acanthogobius ommaturus*	Gut, gill, liver, and muscle	Fresh	-	2.4 MPs individuals^−1^	86.3%	[48]
	*Boleophthalmus pectinirostris*				3.3 MPs individuals^−1^	78.0%	
	*Coilia ectenes*				0.8 MPs individuals^−1^	50.0%	
	*Coilia mystus*				0.3 MPs individuals^−1^	27.5%	
	*Collichthys lucidus*				1.4 MPs individuals^−1^	52.3%	
	*Cynoglossus robustus*				0.8 MPs individuals^−1^	55.5%	
	*Harpodon nehereus*				1.7 MPs individuals^−1^	64.0%	
	*Hemibarbus maculatus*				0.9 MPs individuals^−1^	61.0%	
	*Liza haematocheila*				1.4 MPs individuals^−1^	51.0%	
	*Pampus cinereus*				1.0 MPs individuals^−1^	67.0%	
	*Scomber japoicus*				1.6 MPs individuals^−1^	67.0%	
	*Thamnaconus septentrionalis*				0.65 MPs individuals^−1^	27.5%	
	*Tridentiger barbatus*				2.95 MPs individuals^−1^	78.0%	
Asia/China	*Epinephelus fuscoguttatus × Epinephelus lanceolatus*	Intestine and stomach.		Fiber (70.1%), fragment (23.6%), film (6.1%), and pellet (0.5%).	35.36 MPs individuals^−1^	100%	[49]
Asia/Indonesia	*Decapterus macrosoma*	Gastrointestinal tract	Fresh	Styrofoam and fragment	2.5 ± 6.3 MPs species^−1^	29.4%	[47]
	*Katsuwonus pelamis*			-	0.0 MPs species^−1^	0.0%	
	*Lutjanus gibbus*			-	0.0 ± 0.0 MPs species^−1^	0.0%	
	*Oreochromis niloticus*			-	0.0 ± 0.0 MPs species^−1^	0.0%	
	*Rastrelliger kanagurta*			Fragment, film, and monofilament	1.0 ± 1.1 MPs species^−1^	55.6%	
	*Selar boops*			-	0.0 ± 0.0 MPs species^−1^	0.0%	
	*Siganus argenteus*			Fragment	0.5 ±0.7 MPs species^−1^	50.0%	
	*Siganus canaliculatus*			Monofilament	0.3 ± 0.7 MPs species^−1^	33.3%	
	*Siganus fuscescens*			-	0.0 ± 0.0 MPs species^−1^	0.0%	
	*Spratelloides gracilis*			Fragment	1.1 ± 1.7 MPs species^−1^	40.0%	
Asia/Malaysia	*Clarias gariepinus*	Gills and viscera, including digestive tract	Fresh	Fragment (67.4%), fibre (16.3%), and film (16.3%)	9.0 MPs species^−1^	60%	[113]
	*Colossoma macropomum*				5.0 MPs species^−1^	40%	
	*Ctenopharyngodon idella*				4.0 MPs species^−1^	30%	
	*Epinephelus coioides*				4.0 MPs species^−1^	40%	
	*Euthynnus affinis*				0.0 MPs species^−1^	0%	
	*Eleutheronema tridactylum*				10.0 MPs species^−1^	40%	
	*Megalaspis cordyla*				2.0 MPs species^−1^	20%	
	*Nemipterus bipunctatus*				1.0 MPs species^−1^	10%	
	*Rastrelliger kanagurta*				5.0 MPs species^−1^	50%	
	*Selar boops*				0.0 MPs species^−1^	0%	
	*Thunnus tonggol*				3.0 MPs species^−1^	20%	
Asia/Malaysia	*Atule mate*	Gut	Fresh	Fragment (49.5%), fiber (41.9%), pellet (7.6%), and film (0.9%)	6.3 ± 4.9 MPs individuals^−1^	100%	[52]
	*Crenemugil seheli*				5.0 ± 3.7 MPs individuals^−1^	100%	
	*Rastrelliger brachysoma*				6.2 ± 3.3 MPs individuals^−1^	100%	
	*Sardinella fimbriata*				6.5 ± 4.3 MPs individuals^−1^	100%	
Asia/Saudi Arabia	*Acanthopagrus catenula*	Gills and muscles	Fresh	Fiber and particle	4.0 MPs species^−1^	100%	[46]
	*Calotomus viridescens*			Fiber and particle	9.0 MPs species^−1^	100%	
	*Caranx caninus*			Particle	2.0 MPs species^−1^	100%	
	*Chanos chanos*			Particle	5.0 MPs species^−1^	100%	
	*Centropristis striata*			Particle	5.0 MPs species^−1^	100%	
	*Epinephelus morio*			Particle	5.0 MPs species^−1^	100%	
	*Hemiramphus far*			Particle	6.0 MPs species^−1^	100%	
	*Lethrinus nebulosus*			Particle	6.0 MPs species^−1^	100%	
	*Mullus barbatus*			Particle	3.0 MPs species^−1^	100%	
	*Mugil cephalus*			Fiber and particle	6.0 MPs species^−1^	100%	
	*Netuma thalassina*			Particle	6.0 MPs species^−1^	100%	
	*Oreochromis spilurus*			Particle	2.0 MPs species^−1^	100%	
	*Pagrus major*			Particle	6.0 MPs species^−1^	100%	
	*Pampus argenteus*			Particle	7.0 MPs species^−1^	100%	
	*Pomadasys argenteus*			Particle	2.0 MPs species^−1^	100%	
	*Sardina pilchardus*			Fiber	1.0 MPs species^−1^	100%	
	*Scomber scombrus*			Particle	9.0 MPs species^−1^	100%	
	*Sphyraena barracuda*			Particle	7.0 MPs species^−1^	100%	
	*Saurida undosquamis*			Particle	6.0 MPs species^−1^	100%	
	*Squalus acanthias*			Particle	6.0 MPs species^−1^	100%	
	*Thunnus orientalis*			Particle	10.0 MPs species^−1^	100%	
	*Trachurus indicus*			Fiber and particle	3.0 MPs species^−1^	100%	
Asia/Thailand	*Caesio cuning*	Gastrointestinal tract	Fresh	Fiber and fragment	0.09 MPs species^−1^	8.7%	[114]
	*Alepes djedaba*				0.10 MPs species ^−1^	10.0%	
	*Atule mate*				0.11 MPs species^−1^	11.4%	
	*Elates ransonnettii*				0.06 MPs species^−1^	5.8%	
	*Eubleekeria splendens*				0.09 MPs species^−1^	8.9%	
	*Nemipterus hexodon*				0.07 MPs species^−1^	6.9%	
	*Rastrelliger kanagurta*				0.40 MPs species^−1^	20%	
	*Sardinella gibbosa*				0.29 MPs species^−1^	14.3%	
	*Saurida undosquami*				0.09 MPs species^−1^	9.4%	
	*Scolopsis taenioptera*				0.0 MPs species ^−1^	0%	
	*Selar crumenophthalmus*				0.18 MPs species^−1^	18.2%	
	*Selaroides leptolepis*				0.05 MPs species^−1^	5.0%	
	*Siganus canaliculatus*				0.03 MPs species^−1^	3.3%	
	*Terapon theraps*				0.0 MPs species^−1^	0%	
	*Upeneus vittatus*				0.22 MPs species^−1^	18.8%	
Australia	*Sardinops neopilchardus*	Gut, skin, and muscle	Fresh	-	2.9 mg g^−1^	90%	[56]
Australia	*Arripis georgianus*	Gastrointestinal tract	Fresh	Fiber (81.8%), fragment (12.7%), and film (5.5%).	0.60 ± 0.18 MPs species^−1^	30.0%	[54]
	*Arripis trutta*				1.60 ± 0.76 MPs species^−1^	43.0%	
	*Chrysophrys* *auratus*				1.05 ± 0.26 MPs species^−1^	30.7%	
	*Hyporhamphus* *melanochir*				0.27 ± 0.05 MPs species^−1^	23.3%	
	*Mugil cephalus*				0.94 ± 0.18 MPs species^−1^	50.0%	
	*Platycephalus* *fuscus*				1.14 ± 0.44 MPs species^−1^	42.9%	
	*Platycephalus* *richardsoni*				0.56 ± 0.13 MPs species^−1^	33.3%	
	*Sardinops sagax*				0.32 ± 0.15 MPs species^−1^	14.3%	
	*Sillaginodes* *punctatus*				1.60 ± 0.21 MPs species^−1^	50.3%	
Western America	*Atherinopsis californiensis*	Gastrointestinal tract	Fresh	Fiber	0.6 ± 0.9 MPs species^−1^	33.3%	[47]
	*Citharichthys sordidus*			Fiber and film	1.0 ± 1.2 MPs species^−1^	60.0%	
	*Engraulis mordax*			Fiber and fragment	1.6 ± 3.7 MPs species^−1^	30.0%	
	*Morone saxatili*			Fiber, film, and foam	0.9 ± 1.2 MPs species^−1^	28.6%	
	*Oncorhynchus tshawytscha*			Fiber	0.25 ± 0.5 MPs species^−1^	25.0%	
	*Ophiodon elongatus*			Film	1.0 ± 0.3 MPs species^−1^	9.0%	
	*Scomber japonicus*			-	0.0 ± 0.0 MPs species^−1^	0.0%	
	*Sebastes caurinus*			-	0.0 ± 0.0 MPs species^−1^	0.0%	
	*Sebastes flavidus*			Fiber	0.3 ± 0.6 MPs species^−1^	33.3%	
	*Sebastes miniatus*			-	0.0 ± 0.0 MPs species^−1^	0.0%	
	*Sebastes mystinus*			Fiber	0.2 ± 0.4 MPs species^−1^	20.0%	
	*Thunnus alalunga*			-	0.0 ± 0.0 MPs species^−1^	0.0%	

**Table 3 foods-11-03976-t003:** Road map highlighting the known facts and future directions towards filling research gaps.

Present Scenario	Future Recommendations
Many reports on evidence of microparticles in fish/shrimp/crabs/seaweeds exist	Need to evaluate fate of MPs within fishes because of possibility of translocation from gut to edible tissues
Very few reports on effect on human consumption	Need to evaluate life cycle assessment of MPs using laboratory-based pilot studies. Need to study impact of MP-polluted seafood on human health
Rarer reports on effect on animals feeding on MPs-polluted fish	Plan mitigation measures to reduce/avoid MPs-based impacts

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
