# Peer review of "Overviewing the Ground Reality of Microplastic Effects on Seafoods, Including Fish, Shrimps and Crabs: Future Research Directions"

_foods, 2022, doi:10.3390/foods11243976_

Round 1

Reviewer 1 Report

This study is generally well written, but the title “Overviewing the status quo of microplastics in seafoods like fish, shrimps and crabs, the ground reality: How egregious is the scenario?” is apparently overstated. Here are my major concerns:

[1] The authors provided a data summary of selected papers, but it seems that a lot of relevant papers are missing and the number of included papers may not be sufficient to provide an “overview of the status quo”. How did the authors decide which papers to be included in the review?

[2] The data summary is provided for fishes (Table 2), but not for “shrimps and crabs” claimed in the title.

[3] Table 2 is not easy for data comparison. I would suggest the authors using bar charts or other charts to summarise the data.

[4] More importantly, the question “how egregious is the scenario” asked in the title remains unanswered. It is important for a review to reanalyse available data and to get its own conclusions.

[5] Figure 1 and Figure 2 are confusing that should be redone. It is not clear what messages to be provided from the figures. Where did the authors obtain the photos and images from? Is there any copyright issue?

[6] Fig. 3 should be a table, rather than a figure.

Author Response

This study is generally well written, but the title “Overviewing the status quo of microplastics in seafoods like fish, shrimps and crabs, the ground reality: How egregious is the scenario?” is apparently overstated. Here are my major concerns:

Ans. We would like to thank the Editor and Reviewers for their time and efforts on our manuscript. We thank you for the revision opportunity. We have now revised the manuscript according to your comments. We have also given a point by point response to the queries raised. Thank you.

About your concern on the title , we have now modified the title, to make it sound more subtle. Thank you.

[1] The authors provided a data summary of selected papers, but it seems that a lot of relevant papers are missing and the number of included papers may not be sufficient to provide an “overview of the status quo”. How did the authors decide which papers to be included in the review?

 Ans. We had shortlisted all the papers that deal with fish shrimp and crabs and microplastics. Based on this we have consolidated the reports that have been published on these aspects. We have now further searched and added what was possible. Thank you.

[2] The data summary is provided for fishes (Table 2), but not for “shrimps and crabs” claimed in the title.

Ans. Table 2 as such as you could see as such is elaborate running pages. We chose to address the most popular aspect that has been extensively studied in detail. Moreover the table represents a region wise survey of microplastic reports on fishes. Thank you for your understanding.

[3] Table 2 is not easy for data comparison. I would suggest the authors using bar charts or other charts to summarise the data.

Ans. We would like to explain that Table 2 as we pointed out, is an extensive compilation of a lot of data we accumulated right across decades of publications, where we provide details on microplastic shape, quantity, regional distinction, rate of occurrence, this will be impossible to reflect in a bar diagram. Thank you for your kind understanding.

[4] More importantly, the question “how egregious is the scenario” asked in the title remains unanswered. It is important for a review to reanalyse available data and to get its own conclusions.

Ans. Yes we completely agree, we have now discussed this in the final section. Thank you for your valuable comment. Appreciate that.

[5] Figure 1 and Figure 2 are confusing that should be redone. It is not clear what messages to be provided from the figures. Where did the authors obtain the photos and images from? Is there any copyright issue?

 Ans. About Figure 1 and 2, they were drawn using Canva software. We have a premium account in it. No copyright issues involved. No worries. We have described the figures in the figure captions, to resolve the confusion. Thank you.

[6] Fig. 3 should be a table, rather than a figure.

Ans. Yes, sure, we have now changed that to a Table as Table 3. Thank you very much for your valuable suggestions.

Reviewer 2 Report

Very well-structured review with sharp focus. The article is well written and poses relevant research questions. I have some observations to strengthen further the paper:

Section 2.1.1 appears to be shrunken as compared to other world areas; yet, south America is of crucial importance in socioeconomic and environmental studies including of course MPs pollution and their impact on aquatic organisms and food chain. I suggest integrating the up-to-date-literature on MP ingestion with recent studies, see for instance:

https://doi.org/10.1016/j.marpolbul.2020.111799

https://doi.org/10.1016/j.marpolbul.2021.112371

https://doi.org/10.1007/s00128-019-02604-2

https://doi.org/10.1016/j.envpol.2016.11.

The authors discuss the argument related to the presence of MPs in the guts of fishes/shrimps, where the focus is on entrail removal prior to consumption vs possible migration of MPs in other parts of the edible organisms. I think the authors should add a crucial piece of discussion on the additives/plasticizers/dyes contained in MPs; these compounds would migrate in the edible organisms even if the majority of MPs fragment remain in the entrail and digestive parts. Such additives can be toxic for the fishes/shrimps as well as for humans and other animals that feed on the fishes. This aspect should be thoroughly discussed and added to the review, and keywords such as “additives”, “dyes”, “phthalates”, “heavy metals”, and such, should be present throughout the text.

Finally, a small paragraph could be dedicated to artificial cellulose fibers (textiles, rayon). Not strictly MPs, but they have been demonstrated to participate in pollution as they can contain toxic additives (see above), and their ingestion by fishes has been demonstrated in the literature, the authors can easily find relevant studies on this topic.

Author Response

Very well-structured review with sharp focus. The article is well written and poses relevant research questions. I have some observations to strengthen further the paper:

Ans. At the onset, would like to thank the reviewer for the encouraging and appreciative words, feel so motivated. Thank you. Thank you for your observations, we have now revised as per your suggestions

Section 2.1.1 appears to be shrunken as compared to other world areas; yet, south America is of crucial importance in socioeconomic and environmental studies including of course MPs pollution and their impact on aquatic organisms and food chain. I suggest integrating the up-to-date-literature on MP ingestion with recent studies, see for instance:

https://doi.org/10.1016/j.marpolbul.2020.111799

https://doi.org/10.1016/j.marpolbul.2021.112371

https://doi.org/10.1007/s00128-019-02604-2

https://doi.org/10.1016/j.envpol.2016.11.

Ans. We have now added onto sec 2.1.1. also added the references you have cited, thank you.

The authors discuss the argument related to the presence of MPs in the guts of fishes/shrimps, where the focus is on entrail removal prior to consumption vs possible migration of MPs in other parts of the edible organisms. I think the authors should add a crucial piece of discussion on the additives/plasticizers/dyes contained in MPs; these compounds would migrate in the edible organisms even if the majority of MPs fragment remain in the entrail and digestive parts. Such additives can be toxic for the fishes/shrimps as well as for humans and other animals that feed on the fishes. This aspect should be thoroughly discussed and added to the review, and keywords such as “additives”, “dyes”, “phthalates”, “heavy metals”, and such, should be present throughout the text.

Ans. Yes you are right, actually there are no clear reports with respect to this issue, we have now added a section highlighting this lacunae in the future perspective. Thank you, for these useful suggestions.

Finally, a small paragraph could be dedicated to artificial cellulose fibers (textiles, rayon). Not strictly MPs, but they have been demonstrated to participate in pollution as they can contain toxic additives (see above), and their ingestion by fishes has been demonstrated in the literature, the authors can easily find relevant studies on this topic.

Ans. Since this is not actually directly related to the review, we have mentioned it with an added reference. Thank you again for your time and timely review.

Round 2

Reviewer 1 Report

The authors have addressed my concerns.